# Synapsin II Directly Suppresses Epileptic Seizures In Vivo

**DOI:** 10.3390/brainsci12030325

**Published:** 2022-02-28

**Authors:** Ryan Schwark, Rodrigo Andrade, Maria Bykhovskaia

**Affiliations:** 1Department of Neurology, Wayne State University School of Medicine, Detroit, MI 48203, USA; rs4192@columbia.edu; 2The Zuckerman Institute, Columbia University, New York, NY 10027, USA; 3Department of Pharmacology, Wayne State University School of Medicine, Detroit, MI 48203, USA; randrade@med.wayne.edu

**Keywords:** synapse, AAV, epileptiform, EPSC, 4-AP, PKA

## Abstract

The synapsin family offers a strong linkage between synaptic mechanisms and the epileptic phenotype. Synapsins are phosphoproteins reversibly associated with synaptic vesicles. Synapsin deficiency can cause epilepsy in humans, and synapsin II (SynII) in knockout (KO) mice causes generalized epileptic seizures. To differentiate between the direct effect of SynII versus its secondary adaptations, we used neonatal intracerebroventricular injections of the adeno-associated virus (AAV) expressing SynII. We found that SynII reintroduction diminished the enhanced synaptic activity in *Syn2* KO hippocampal slices. Next, we employed the epileptogenic agent 4-aminopyridine (4-AP) and found that SynII reintroduction completely rescued the epileptiform activity observed in *Syn2* KO slices upon 4-AP application. Finally, we developed a protocol to provoke behavioral seizures in young *Syn2* KO animals and found that SynII reintroduction balances the behavioral seizures. To elucidate the mechanisms through which SynII suppresses hyperexcitability, we injected the phospho-incompetent version of *Syn2* that had the mutated protein kinase A (PKA) phosphorylation site. The introduction of the phospho-incompetent SynII mutant suppressed the epileptiform and seizure activity in *Syn2* KO mice, but not to the extent observed upon the reintroduction of native SynII. These findings show that SynII can directly suppress seizure activity and that PKA phosphorylation contributes to this function.

## 1. Introduction

Epilepsy is a complex and multifaceted disorder, and it has been established that changes in the presynaptic function can contribute to the etiology of epilepsy [1]. In particular, some forms of epilepsy have been linked to deficiencies in synapsin function [2].

The synapsins are a family of neuronal phosphoproteins abundantly expressed in the brain [3]. They are encoded by three genes (*Syn1*, *Syn2,* and *Syn3*) [4] and are concentrated in synaptic terminals. Within synaptic terminals, they reversibly associate with synaptic vesicles and play key roles in vesicle clustering and mobilization for the release process [5,6,7,8]. The synapsin function depends on its phosphorylated state, which is regulated by multiple kinases and phosphatases [6]. In particular, synapsin binding to and dissociation from vesicles are mediated through phosphorylation at a highly conserved site P1 [6], a target for protein kinase A (PKA) and calcium/calmodulin kinase I (CaMKI) [9].

Evidence that the synapsins can contribute to the pathophysiology of epilepsy emerged from the observation that knockout (KO) mice lacking Syn I or Syn II exhibited a robust epileptic phenotype [10,11]. Subsequently, an *Syn1* nonsense mutation was found to be associated with familial epilepsy, autism, and mental retardation [12]. Further studies identified additional nonsense and missense mutations in *Syn1* associated with intellectual disabilities [13,14], autism, and epilepsy [13,15], thus supporting a role for synapsins in human epilepsies. Finally, a large-scale search for genetic susceptibility loci in epilepsy identified *Syn2* as one of the five major genes that contribute to epilepsy predisposition [16]. These findings were supported by subsequent studies revealing that polymorphism in *Syn2* is associated with idiopathic generalized epilepsy [17,18].

While both *Syn1* and *Syn2* KO mice exhibit seizures, the *Syn2* KO animals show a more robust epileptic phenotype consisting of tonic-clonic and running generalized seizures that start during early adolescence [19,20,21]. These observations point to the *Syn2* KO mice as an ideal animal to study synapsin-related epilepsies.

Although synapsin function has been extensively studied, the synapsin functions in nerve terminals contributing to overexcitability in synapsin deficient brains and what the relative contribution of the direct effect of synapsin-controlled mechanisms versus compensatory adjustments in neuronal networks are still debated [22]. To elucidate these questions, we investigated how reintroduction of Syn II affects overexcitability in *Syn2* KO brains. We built on recent work showing that adeno-associated viruses (AAVs) effectively diffuse from the cerebral ventricles into the brain while the ependymal lining is still immature [23,24,25]. This property made it possible to use intracerebroventricular (ICV) injection of AAVs to express Syn II in the *Syn2* KO brain in vivo.

## 2. Materials and Methods

### 2.1. Animals

Mice heterozygous for the *Syn2* targeted mutation were purchased from The Jackson Laboratory (strains B6;129S-*Syn2^tm1Sud^*/J and B6). Homozygote *Syn2* KO and wild-type (WT) lines were derived from breeding heterozygotes. The genotypes of all of the breeders were independently confirmed by a commercial genotyping service (Transnetyx). All the animal colonies were kept at standard conditions at room temperature and on a 12 h dark/light cycle. The husbandry conditions were identical for all the strains in the study. All the experiments were performed in accordance with the guidelines of the Animal Care and Use Committee (IACUC) of the Wayne State University School of Medicine (WSU SOM) and the National Institutes of Health of the US Public Health Service. All the procedures were approved by the WSU SOM IACUC (Protocol # 18-03-0592).

### 2.2. AAV Injections

An adeno-associated virus (AAV1.hSyn.Syn2-Turbo-GFP.WPRE.hGh) was custom-made by Penn Vector Core to contain AAV serotype 1 and an *hSyn* promoter to drive the expression of GFP-tagged *Syn2* in neurons [1,2] with WPRE and hGh poly A sequences. The GFP-tagged *Syn2* (variant IIa) vector was commercially obtained from Origene (CAT#: MG225317). The AAV (1-2 μL, ~10^9^–10^10^ gc/mouse) was injected into the lateral ventricle(s) of newborn (P0-P1) *Syn2* KO mice as described by Kim et al. [24].

### 2.3. Slice Preparation

Mice (15 to 20 days old) were anesthetized by isoflurane and sacrificed by decapitation. Their brains were removed rapidly and bathed in an ice-cold ACSF solution containing (in mM): 120 NaCl, 2.5 KCl, 1.3 MgSO_4_, 1.0 KH_2_PO_4_, 2.5 CaCl_2_, 24.0 NaHCO_3_, 10 dextrose, and 9 sucrose, with the osmolality adjusted to 295 mOsm with sucrose, and bubbled with a mixture of 95% O_2_ and 5% CO_2_. Horizontal hippocampal slices (300 µm thickness) were cut using a vibratome Leica VT 1200 S. The slices were washed two times in ACSF solution (previously bubbled with the same gas mixture), and maintained in the recording solution continuously bubbled with the same gas mixture for one hour prior to the experiment.

### 2.4. Electrophysiology

Slices were submerged in the recording chamber and were continuously superfused with the recording solution. Whole-cell recordings were performed from hippocampal CA1 neurons. Recordings were acquired using a multiclamp 700B amplifier and PClamp 10.0 software (Axon Instruments) and digitized at 10 KHz. Recordings of spontaneous and epileptiform activity were performed using a potassium-gluconate-based intracellular solution (Luhman et al., 2000) containing (in mM): 135 K-gluconate, 0.1 CaCl_2_, 2.0 MgCl_2_, 2.0 Na-ATP, 1.0 EGTA, and 10 HEPES, at pH 7.35 (adjusted with KOH), 280 mOsm. Under these recording conditions, E_Cl_ was −90 mV and cells were held at −50 mV to allow a clear distinction between spontaneous excitatory and inhibitory postsynaptic currents (sEPSCs and sIPSCs, respectively). N-methyl-D-aspartate (NMDA) type glutamatergic receptors were blocked by D-(-)-2-amino-5-phosphonopentanoic acid (D-AP5 20 µM). 4-aminopyridine (4-AP, 30 µM, Sigma) was applied to evoke epileptiform activity.

### 2.5. Immunohistochemistry

Mice were perfused with 4% paraformaldehyde in phosphate-buffered saline (PBS), and their brains were removed and then postfixed overnight. Coronal or horizontal sections containing the hippocampus were sectioned at 50 µm using a vibratome and processed for immunolabeling as described in [26]. In some cases, we conducted immunolocalization of GFP and Syn II in slices sectioned for electrophysiology (300 µm nominal thickness). In these case, slices were fixed in 4% paraformaldehyde in PBS for 0.5–4 h and processed for immunolocalization, as performed when using thinner slices, except that the antibody incubation periods were increased to 2–3 days with gentle agitations at 4 °C. The primary antibodies (1:1000 dilution) used for these experiments were rabbit polyclonal anti-Syn II (Abcam ab13258) and chicken anti-GFP (Aves Labs). The secondary antibodies (1:100 dilution) were goat anti-rabbit IgG conjugated to Rhodamine Red and goat anti-Chicken IgG conjugated to Alexa Fluor 488 or Dylight 488 (all from Jackson Immunoresearch). In some cases, slices were stained using Neurotrace 500/525 (Fluorescent Nissl) after immunostaining. The slices were extensively rinsed with PBS, incubated in Neurotrace 500/525 (1:100 in PBS) for 30 min with gentle agitation, and then extensively rinsed. All images were acquired using an Olympus Fluoview confocal microscope with identical settings. The image analysis was performed using ImageJ (National Institute of Health).

### 2.6. Behavioral Seizures

Seizures were provoked by lifting mice by the tail, since earlier studies showed that this protocol robustly induces generalized tonic-clonic seizures in *Syn2* KO mice [19,20,21,27], resulting in seizures typically involving forelimb myoclonus followed by hindlimb myoclonus/clonus and truncus tonic-clonic activity. To quantify seizure activity, we videotaped the mice over the course of 3 min tail suspensions (TS) [28] and determined the percentage of time displaying seizures. A seizure was defined as a truncal element (epistotonus, emprostotonus, or emprostoclonus) accompanied by fore- and hindlimb myoclonus or clonus.

### 2.7. Data Analysis

Video recordings were analyzed using Lightworks software (LWKS). The electrical recordings were analyzed using Clampfit software (Molecular Devices) and in-house QUANTAN software [29]. Images were processed using FIJI/ImageJ (National Institute of Health). ANOVA followed by the Tukey post hoc test was employed for multiple comparisons.

## 3. Results

The aim of the present work was to re-express Syn II into the brains of *Syn2* KO newborn mice to assess the ability of this reintroduction to rescue the cellular and whole-animal epileptic phenotype. Previous work showed that it is possible to tag Syn II with GFP without impacting the function of this protein [30]; therefore, we expressed Syn II as a GFP fusion protein to allow for routine assessment of the infection’s spread. In some experiments, we complemented this approach by direct immunolocalization of Syn II. To implement this second approach, we first validated Syn II immunolocalization by comparing the immnolocalization obtained in slices derived from WT and *Syn 2* KO mice. WT and *Syn2* KO mouse brains were perfused, sectioned, and processed for immunohistochemistry in parallel using a common set of reagents and solutions. We observed robust punctate staining of Syn II in slices derived from WT mice but only low-level, diffuse (i.e., background) staining in slices derived from *Syn2* KO mice (Figure 1). These findings support the feasibility of assessing expressions of Syn II over the null background.

Next, we injected AAV1.hSyn.Syn2-Turbo-GFP in the lateral ventricles of neonatal *Syn2* KO mice and assessed Syn II-GFP expression at P20 by examining GFP fluorescence and Syn II expression. As illustrated in Figure 2A, neonatal ICV injection of the AAV expressing GFP-tagged Syn II resulted in robust expression of GFP in the cortex and hippocampus two weeks after the injection (the earliest time point assessed). Importantly, expression of the GFP tag was colocalized. Notably, GFP and Syn II coimmunolocalized (Figure 2B,C), confirming the robust Syn II expression.

Next, we tested whether the ICV AAV injections would rescue misregulations in synaptic transmission in *Syn2* KO mice. It was previously shown that during the early postnatal period (P13-18) the sEPSC is significantly elevated in *Syn2* KO hippocampal slices [31]. To test whether the reintroduction of Syn II would rescue this defect, we performed ICV injections of AAV1.hSyn.Syn2-Turbo-GFP in *Syn2* KO neonatal mice, and then monitored spontaneous synaptic transmission at P13-P18. In agreement with the findings of an earlier study [31], Syn2 KO mice had an almost two-fold increase in sEPSC frequency (0.98 ± 0.10 Hz at *Syn2* KO versus 0.52±0.04 Hz at WT, *n* = 14). Notably, the *Syn2* KO mice injected with AAV1.hSyn.Syn2-Turbo-GFP had the sEPSC frequency reduced to the WT levels (0.56 ± 0.08, Figure 3). This result demonstrated that neonatal expression of Syn II completely rescued the enhanced spontaneous synaptic activity in *Syn2* KO hippocampal slices.

Next, we investigated how the epileptogenic effects of low-dose 4-AP are affected by Syn II re-expression. We built upon an earlier study [32], which employed (4-AP) to investigate ictal and interictal (I-IC) discharges in the brains of synapsin-deleted mice. We previously showed [31] that *Syn2* KO hippocampal slices have an enhanced responsiveness to low concentrations of 4-AP.

4-AP (30 µM) was added to the ACSF extracellular solution. After the baseline recordings of spontaneous activity were performed for five minutes, the slices were continuously perfused with the solution containing 4-AP. The recordings of 4-AP-induced activity started after four minutes of perfusion and were performed for five minutes. I-IC events were defined as downward currents with an amplitude exceeding 60 pA and a rapid rising phase (below 1 ms). Such events were totally absent in the absence of 4-AP treatment and were only detected upon 4-AP application.

Consistent with these earlier findings, we observed that administration of 30 μM 4-AP induced robust I-IC discharges in *Syn2* KO but not in WT mice (I-IC frequency of 0.080 ± 0.030 Hz at *Syn2* KO versus 0.007 ± 0.006 Hz at WT, *n* = 9). Notably, the 4-AP induced I-IC frequency in the *Syn2* KO animals injected with AAV1.hSyn.Syn2-Turbo-GFP was reduced to the WT levels (Figure 4). Thus, the AAV-mediated neonatal expression of *Syn2* rescued the resistance to the epileptogenic effects of low-dose 4-AP. These results demonstrate that the reintroduction of Syn II during the early postnatal period rescued the cellular hyperexcitable phenotypes of the *Syn2* KO mice.

Next, we tested whether the reintroduction of Syn II could rescue seizure activity in *Syn2* KO animals in vivo. We built upon previous studies characterizing seizure activity in *Syn2* KO mice [19,20,21,33,34,35]. These studies have shown that the seizure progression typically involves forelimb myoclonus, followed by hindlimb myoclonus/clonus and truncus tonic-clonic activity (epistotonus, emprostotonus, and emprostoclonus), which is sometimes followed by a running fit. Since it was shown that seizures in *Syn2* KO adult mice can be provoked by gently lifting a mouse by the tail [20,21,27], we employed the tail suspension (TS) method [28]. Each TS mouse was individually videotaped over the course of 3 min. We then analyzed the videorecordings and quantified the percentage of time seizure behaviors were observed. A seizure was defined as a truncal element (epistotonus, emprostotonus, or emprostoclonus) accompanied by fore-/hindlimb myoclonus or clonus.

Notably, we found that this protocol provoked seizures even in young (P15) *Syn2* KO animals (Appendix A), which was not observed earlier. The *Syn2* KO mice at P15 showed seizures during 64.8% ± 14.6% of their TS time. In contrast, seizure elements were completely absent in WT mice at P15 (Appendix A). This result shows that young *Syn2* KO animals, which were previously considered asymptomatic, can demonstrate seizure behaviors when consistently provoked.

In agreement with earlier studies [19,20,21,33,34], we observed age-dependent seizure enhancement in *Syn2* KO mice (Figure 5A, red). At the age of three months, the *Syn2* KO mice exhibited seizures continuously (100% of time) over the entire course of the TS. In contrast, adult WT mice showed seizure elements only occasionally (10.2% ± 5.1% of their TS time).

We then employed this protocol to investigate seizure activity in *Syn2* AAV-injected *Syn2* KO mice. Notably, at P15, seizure behavior was almost completely rescued by Syn II expression (Figure 5, blue; Appendix A). To ensure that that the observed result was not influenced by the mechanical ICV action, we performed a mock injection of AAV1.hSyn.Turbo-GFP, which did not express Syn II. We found that in contrast to the robust rescue observed upon expression of SynII-GFP, seizure activity following the expression of GFP alone was indistinguishable from the control *Syn2* KO at P15 and significantly different from the Syn II expressing rescue (Figure 5B, the green bar). Interestingly, subsequent developmental time-points (1, 2, and 3 months) showed less pronounced, but still significant, rescue produced by Syn II expression. This result demonstrates that neonatal AAV injections can be used to rescue the epileptic phenotype at the whole-animal behavioral level.

Finally, we investigated how Syn II phosphorylation at the PKA- and CaMKI-dependent site P1, which regulates vesicle clustering (Figure 6A), contributes to the regulation of epileptiform and seizure activity. We generated the AAV incorporating the S → A mutation in Syn II at site P1 (AAV1.hSyn.*Syn2^S10A^*-Turbo-GFP). We next performed the ICV of AAV1.hSyn.*Syn2^S10A^*-Turbo-GFP and investigated the synaptic, epileptiform, and seizure activity in the injected animals (Figure 6). Surprisingly, we found that the sEPSC frequency in the animals injected with mutated *Syn2^S10A^* was similar to that in animals injected with native *Syn2* and was indistinguishable from WT controls (Figure 6A). This result suggests that the dephospho-Syn II form is needed to balance spontaneous synaptic activity, and that Syn II phosphorylation at the P1 site is not required for this Syn II function. However, this was not the case for either 4-AP-induced epileptiform activity recorded from brain slices (Figure 6B) or for behavioral seizure activity (Figure 6C). In both cases, Syn II^S10A^ mutants only produced a partial rescue for epileptic seizures in *Syn2* KO animals and for epileptiforms at *Syn2* KO hippocampal slices. The I-IC frequency induced by 4-AP (Figure 6B) and seizure activity at P14 (Figure 6C) was significantly higher in the animals injected with *Syn2^S10A^* compared to that in the animals injected with *Syn2*. This result suggests that Syn II phosphorylation at site P1 contributes to the ability of Syn II to balance epilepsy predisposition.

## 4. Discussion

It is still debated which mechanisms are responsible for producing overexcitability and epilepsy in the synapsin-deficient brain. The finding that synapsins promote GABAergic inhibitory transmission [36] led to the hypothesis that synapsin deficiency shifts the excitation/inhibition balance in neuronal networks toward excitation [37]. This hypothesis was further supported by the discovery that Syn II maintains the asynchronous component of inhibitory transmission [38], which contributes to balancing seizure activity [39]. Furthermore, at glutamatergic neurons, Syn II regulates Ca^2+^-dependent plasticity [31] and neuronal outgrowth [40], and these mechanisms can also contribute to epilepsy predisposition. Finally, synapsin deletion likely initiates a complex cascade of events, which involve secondary and compensatory changes in brain activity, which can also contribute to the development of epileptic seizures [22,41].

To start unraveling these mechanisms, we asked whether Syn II impacts epileptic activity directly, and investigated the contributions of the direct Syn II action in the synapse versus developmental compensatory mechanisms in the Syn II deleted brain. To address these questions, we employed neonatal ICV injections of AAVs harboring *Syn2*. We first demonstrated that this approach enables broad expression of Syn II in the brain. Importantly, we found that the introduction of Syn II after birth, at P0, balances the hyperexcitability observed in the *Syn2*-deficient brain. As such, these results present the first evidence that the introduction of Syn II in the intact brain in vivo directly suppresses the excitability in neuronal networks and behavioral epileptic seizures.

We showed that seizures can be detected in very young (P14) *Syn2* KO animals when they are consistently provoked, and that seizure activity at this stage can be completely reversed upon Syn II reintroduction at the early postnatal stage (P0–P1). This result provides the first demonstration of the direct Syn II effect on epileptic seizures in vivo in the postnatal brain. As such, it demonstrates that it is the action of Syn II in the presynaptic terminals, as opposed to developmental compensatory mechanisms, that triggers overexcatability in neuronal networks and behavioral seizures.

In should be noted, however, that the neonatal reintroduction of Syn II at P0-P1 produced only a partial rescue of seizure activity in adult animals. Such an incomplete rescue may be due either to the diminished Syn II expression levels or to the compensatory changes in the brain developed in the postnatal period following ICV, which may enhance seizure predisposition in adults. Further experimentation is needed to discriminate between these two possibilities.

We then focused on the next major question: what are the molecular mechanisms underlying the overexcitability in *Syn2* KO brain? Although synapsins have been extensively studied and characterized at the molecular level in vitro and in neuronal cultures [4,5,6,7,8], the exact role of their molecular interactions in vivo is not yet understood. Syn function in synapses is complex and multifaceted [5]. However, it is generally agreed that the major functions of Syn II rely on its ability to reversibly bind synaptic vesicles in a phosphorylation-dependent manner [4,6,7,42], to cluster them, and to direct them to the sites of release [5,43]. Syn II binding to and dissociating from vesicles are mediated through phosphorylation at site P1, a target of protein kinase A (PKA) and calcium/calmodulin kinase I (CaMKI) [9]. This is the only phosphorylation site conserved among all the synapsin isoforms, from invertebrates to vertebrates [6].

To understand the role of the highly conserved site P1 of Syn II in synaptic and epileptic activity, we generated AAV incorporating *Syn^S10A^* phospho-incompetent mutant with the modified P1 site. Notably, injecting the mutated *Syn^S10A^* in the neonatal brain completely reversed the enhanced glutamatergic activity observed in *Syn2* KO brain slices. This result suggests that the supression of spontaneous glutamatergic transmission is mediated by dephospho- Syn II, and the phosphorylation at site P1 does not contribute to this Syn II function.

However, this was not the case for the epileptic activity. Upon the reintroduction of the mutated *Syn^S10A^*, we observed only a partial rescue of behavioral seizures observed in *Syn2* KO animals and of epileptiforms observed at *Syn2* KO brain slices. These results reveal the contribution of PKA/CaMKI phosphorylation of Syn II to the regulation of epilepsy predisposition, even though this mechanism does not apparently contribute to the regulation of spontaneous glutamatergic transmission.

Altogether, our results show that Syn II balances epileptic activity directly and reveals the contribution of PKA/CaMKI phosphorylation to this Syn II action. These initial results also show that the ICV approach can be used to further delineate the mechanisms of multifaceted synapsin function in vivo, for example, by neuron-specific protein reintroduction, which could help identify the neuronal pathway that underlies epilepsy predisposition.

## Figures and Tables

**Figure 1 brainsci-12-00325-f001:**
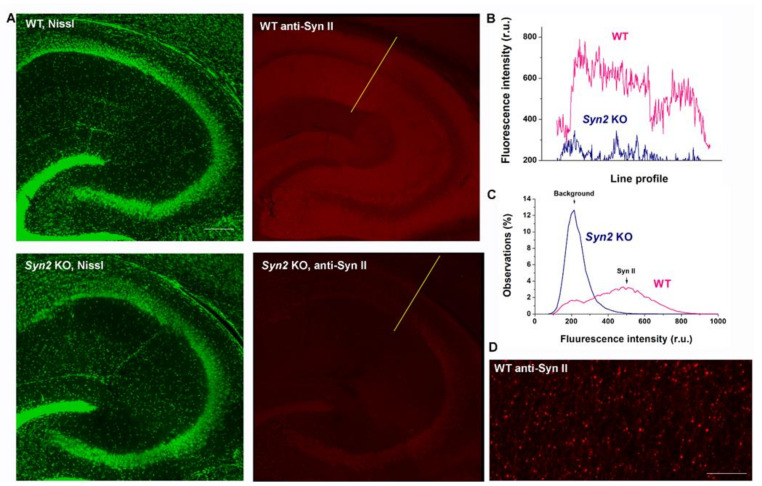
Immunohistochemical detection and quantification of Syn II in the CA1 region of the hippocampus. (**A**) Horizontal slices through the middle hippocampus stained using a fluorescent Nissl (Neurotrace 500/525) to provide an anatomical reference point and immunolabeled for Syn II. Green channel (left): Nissl staining shows similar anatomy for the WT and *Syn2* KO slices. Red channel (right): immunolocalization of Syn II. The left and right panels show identical fields of view. Note a robust Syn II signal detected in the WT slice but only background fluorescence in the *Syn2* KO slice. Scale bars: 200 µm. (**B**) The fluorescence intensity line profiles for the Syn II signal (red channel) over the different cell layers of the CA1 region for these slices as denoted by the yellow lines in panel A. (**C**) The distributions of pixel flurescence intensities for the anti-Syn II signal (red channel) for WT and Syn2 KO slices. Note a single peak (background) for the *Syn2* KO strain and two peaks (background and Syn II signal) for WT. *p* < 0.001 per K.-S. test. (**D**) Syn II immunolabeling has a punctate pattern in WT slices, as expected from Syn II being locatized to nerve terminals. Scale bar: 10 µm.

**Figure 2 brainsci-12-00325-f002:**
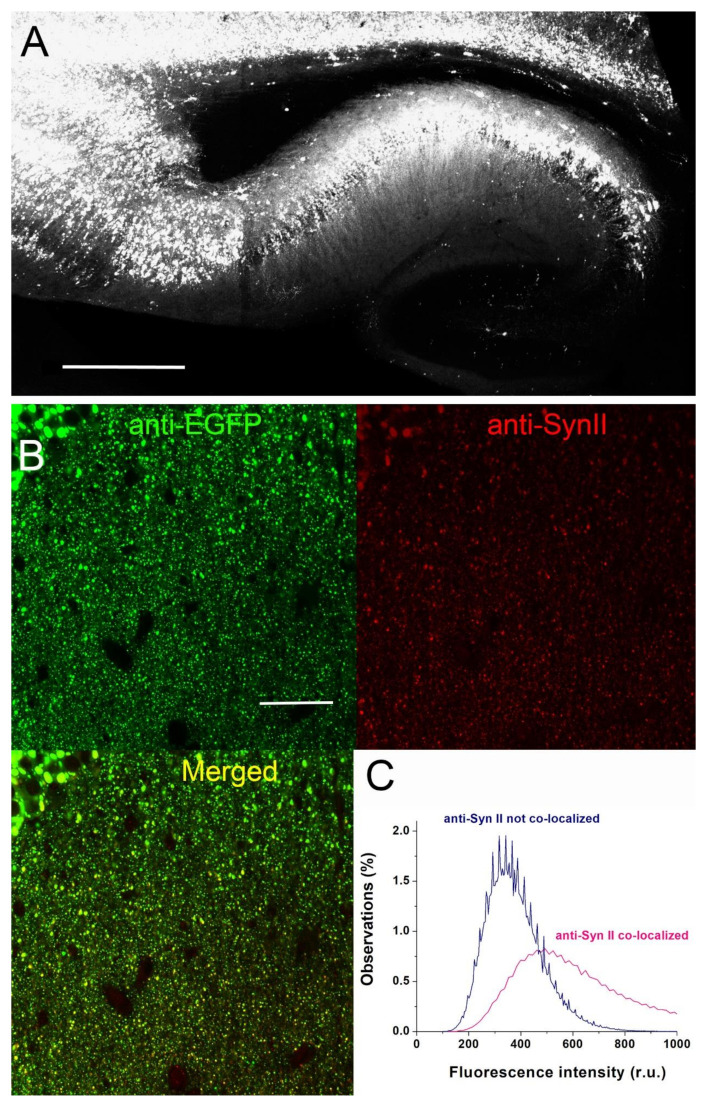
Expression of SynII-TurboGFP in the hippocampus of AAV-injected mice. (**A**) SynII-TurboGFP expression in the hippocampus and cortex after neonatal ICV injection of AAV1.hSyn.Syn2-Turbo-GFP. Scale 500 µm. (**B**) High-magnification image of a small section of the Str. Radiatum showing the punctate immunoreactivity expected for SynII-TurboGFP. Scale 50 µm. (**C**) The distributions of the anti-Syn II fluorescence for the pixels colocalized (pink) and not colocalized (navy) with the anti-GFP signal (green channel). Colocalization was assessed using ImageJ plugin Coloc 2. Note that the non-colocalized signal matches the background peak observed for *Syn2* KO slices (shown in Figure 1C), while the colocalized peak matches the anti-Syn II signal observed in WT slices (*p* < 0.001 per K-S test).

**Figure 3 brainsci-12-00325-f003:**
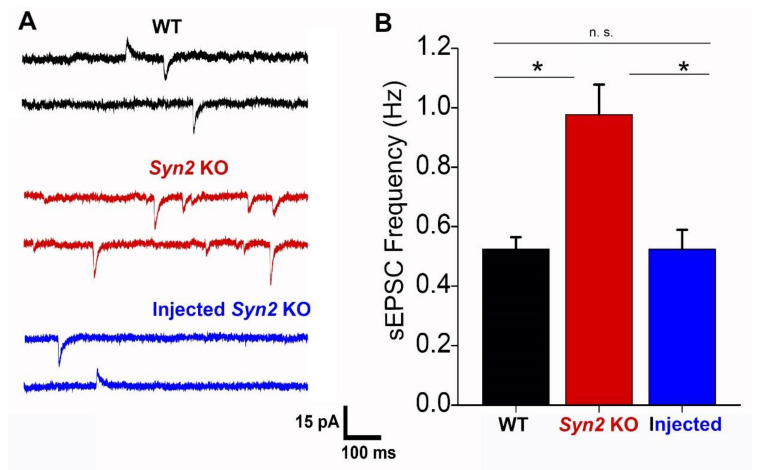
Neonatal expression of Syn II rescues the enhancement in sEPSC frequency observed at *Syn2* KO slices. (**A**) Representative recordings from WT, *Syn2* KO, and AAV1.hSyn.Syn2-Turbo-GFP-injected *Syn2* KO animals. Downward currents correspond to glutamatergic and upward currents correspond to GABAergic synaptic currents (holding potential -50 mV, see [31]). (**B**) ICV injections of AAV1.hSyn.Syn2-Turbo-GFP in *Syn2* KO animals reduced the sEPSC frequency of the WT levels. N = 14 cells (at least 7 animals) per genotype. * *p* < 0.01, per ANOVA followed by Tukey’s test.

**Figure 4 brainsci-12-00325-f004:**
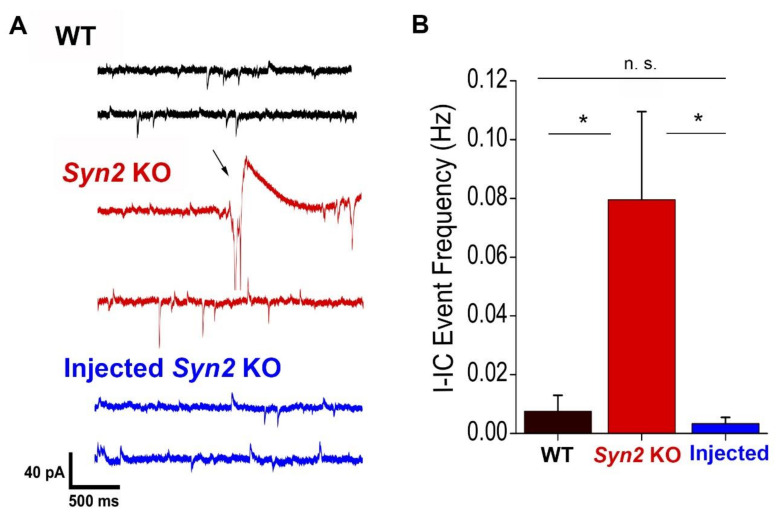
Neonatal *Syn2* expression rescues the epileptiform activity observed upon 4-AP application (30 µM) in *Syn2* KO hippocampal slices. (**A**) Representative recordings from WT, *Syn2* KO, and AAV1.hSyn.Syn2-Turbo-GFP-injected *Syn2* KO animals with 30 µm 4-AP applied to the bath. Arrow denotes an I-IC event [31]. (**B**) I-IC spikes are induced in *Syn2* KO slices upon 4-AP application, and this hyperexcitability is rescued in AAV1.hSyn.Syn2-Turbo-GFP-injected animals. * *p* < 0.05 per ANOVA followed by Tukey’s test, *n* = 9.

**Figure 5 brainsci-12-00325-f005:**
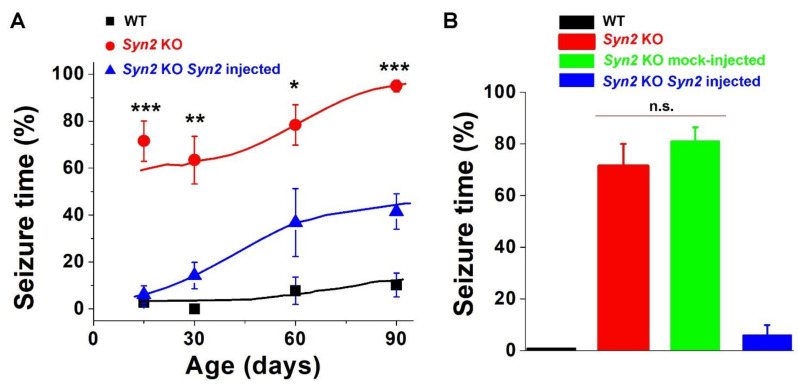
Syn II expression rescues the epileptic phenotype observed in *Syn2* KO animals. (**A**) Seizure activity as a function of age for WT (black), *Syn 2* KO (red), and AAV1.hSyn.Syn2-Turbo-GFP-injected Syn2 KO rescue (blue). The rescue is almost complete at P15 and it weakens as the postnatal age increases. (**B**) Seizure activity at P15 including mock-injected GFP controls (AAV1.hSyn.Turbo-GFP-injected *Syn2* KO). *n* = 9 for each group. * *p* < 0.05, ** *p* < 0.01, and *** *p* < 0.001 per ANCOVA followed by Tukey’s test (injected versus *Syn2* KO).

**Figure 6 brainsci-12-00325-f006:**
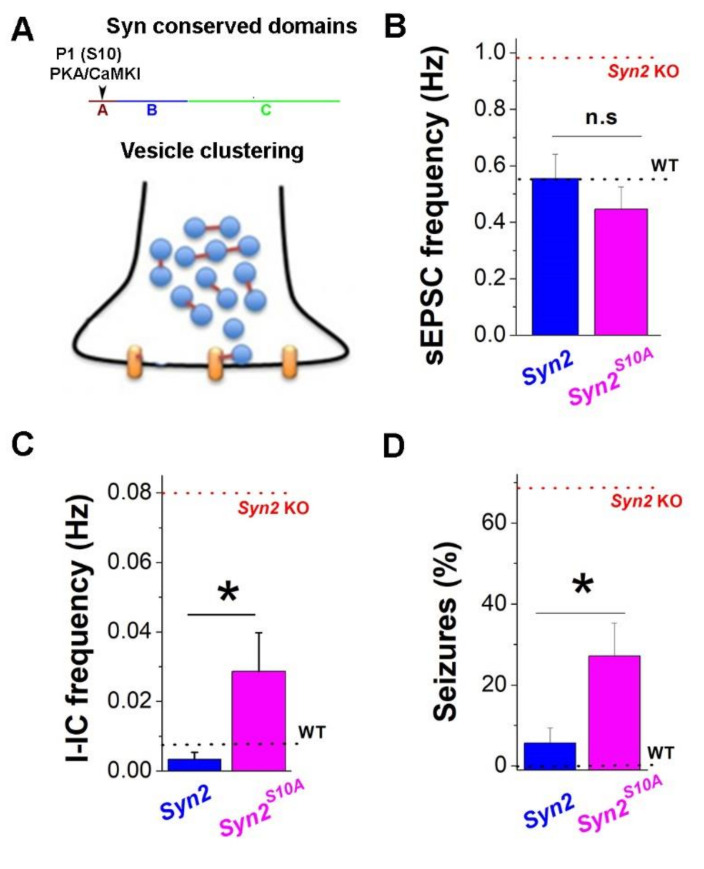
Phosho-incompetent *Syn2^S10A^* mutant partially rescues the overexcitable phenotype in *Syn2* KO animals. (**A**) A diagram showing the conserved domains (**A**–**C**) of Syn II and the PKA/CaMKI phosphorylation site P1 in the A domain. The diagram below illustrates Syn II (red) function in tethering and clustering of synaptic vesicles and directing them to Ca^2+^ channels (yellow). (**B**) ICV of the native *Syn2* and mutant *Syn2^S10A^* genes produce a similar reduction in spontaneous synaptic activity (*n* = 16). (**C**) Expression of mutant *Syn2^S10A^* does not reduce the 4-AP produced epileptiform activity to the extent observed for the expression of native *Syn2* (*p* < 0.05, *n* = 9). (**D**) Expression of mutant *Syn2^S10A^* does not reduce seizure activity to the extent observed for the expression of native *Syn2* (* *p* < 0.05, *n* = 9). Dotted red lines corrrespond to *Syn2* KO animals, and dotted black lines correspond to WT animals (replotted from Figure 3, Figure 4 and Figure 5).

## Data Availability

Data are available on request.

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
