# Peer review of "Synapsin II Directly Suppresses Epileptic Seizures In Vivo"

_brainsci, 2022, doi:10.3390/brainsci12030325_

Round 1
Reviewer 1 Report
Bykhovskaia et al. studied the role of Synapsin II in the generation of seizures. The authors used a Syn2 KO mouse line and reintroduction of two forms of synapsin II (normal and phospho-incompetent version of Syn2) to test the hypothesis about the role of Synapsin II. As a result, new data were obtained showing that SynII can directly suppress seizure activity and that PKA 25 phosphorylation contributes to this function.
Although the authors' conclusions are pretty convincing, the article requires substantial revision. The methodology contains inaccuracies and probably errors that need to be corrected.
- The composition of the extracellular solution (lines 93-94): "120 NaCl, 2.5 KCl, 1.3 MgSO4, 1.0 KH2PO4, 2.5 CaCl2, 24.0 NaHCO3, 10 dextrose, 50 sucrose, with the osmolality adjusted to 295 mOsm with sucrose. Based on the composition, the osmolality of such a solution should be higher. Therefore, probably 50 sucrose is indicated incorrectly.
- It is necessary to describe exactly which solution was used to induce epileptiform activity. Since the result may depend on the exposure time, it is essential to describe in detail how, at what time, and what parameters were recorded.
- The Methods state that recording was done at -50 mV to distinguish between sIPSCs and sEPSCs, but the results refer only to sEPSCs.
- This section needs a complete revision. The primary and secondary antibodies do not match (the primary antibodies (the primary antibodу chicken anti-GFP and the secondary antibody was goat anti-mouse IgG conjugated to Alexa fluor 568. I hope this is only a mistake in the description. The procedure should be described in detail, including a description of how the measurements were taken.
- 1 and Fig. 2 need to be completely redone. The figure itself, the caption under the figures, the description in the Methods and Results do not correspond well. There are no quantitative assessments that are needed to draw a convincing conclusion.
- Fig. 3 and Fig. 4. A control with an injection of a virus containing only GFP is missing. The statistical analysis is not precise. A one-way ANOVA and Tukey's post hoc tests are desirable in this case.
- Figure 6. Missing comparison with controls (WT and virus with GFP)
Author Response
- The composition of the extracellular solution (lines 93-94): "120 NaCl, 2.5 KCl, 1.3 MgSO4, 1.0 KH2PO4, 2.5 CaCl2, 24.0 NaHCO3, 10 dextrose, 50 sucrose, with the osmolality adjusted to 295 mOsm with sucrose. Based on the composition, the osmolality of such a solution should be higher. Therefore, probably 50 sucrose is indicated incorrectly.
Indeed, we made a typo in reporting the sucrose concentration, and we thank the Reviewer for pinpointing this. The concentration of sucrose in our extracellular solutions was 9 mM. This has been corrected in the revised manuscript.
- It is necessary to describe exactly which solution was used to induce epileptiform activity. Since the result may depend on the exposure time, it is essential to describe in detail how, at what time, and what parameters were recorded.
In the revised manuscript (lines 268-274) we described in detail how epileptiform activity was induced and detected.
- The Methods state that recording was done at -50 mV to distinguish between sIPSCs and sEPSCs, but the results refer only to sEPSCs.
Indeed, the holding potential of -50 mV enables discriminating between sIPSCs and sEPSCs and accurate detection of sEPSCs. However, at these conditions the signal to noise ratio for sIPSCs is not sufficient for the accurate sIPSC detection and analysis. To enable accurate detection of sIPSCs, the holding potential needs to be set to 0 mV, as was done in our earlier study (Feliciano et al., J. Neurosci. 2013). However, in the present study we focused on excitatory activity, and therefore the recordings and detection of sIPSCs at the holding potential of 0 mV were not performed.
- This section needs a complete revision. The primary and secondary antibodies do not match (the primary antibodies (the primary antibodу chicken anti-GFP and the secondary antibody was goat anti-mouse IgG conjugated to Alexa fluor 568. I hope this is only a mistake in the description. The procedure should be described in detail, including a description of how the measurements were taken.
Indeed, we made a mistake in the description of the antibodies used in our study, and we thank the Reviewer for pinpointing this. In the revised manuscript we described the immunostaining procedure correctly and in detail.
- 1 and Fig. 2 need to be completely redone. The figure itself, the caption under the figures, the description in the Methods and Results do not correspond well. There are no quantitative assessments that are needed to draw a convincing conclusion.
We revised thoroughly the figures 1 and 2, figure legends, and corresponding text and added the quantitative analysis (Fig. 1 B, C and Fig. 2 C).
- Fig. 3 and Fig. 4. A control with an injection of a virus containing only GFP is missing. The statistical analysis is not precise. A one-way ANOVA and Tukey's post hoc tests are desirable in this case.
The figures 3 and 4 were modified to demonstrate the results of the Tukey’s post hoc tests.
The mock-injection of the AAV containing only GFP was only performed for behavioral experiments. We reasoned that behavioral seizures would be the most sensitive to the mechanical damage caused by injection procedure. However, the figure 5 B clearly demonstrates that the mock-injections had no effect on behavioral seizure activity. Therefore, mock-injections were not performed for the electrophysiology experiments. In these experiments (figures 3 and 4) WT and Syn2 KO slices were used as controls.
- Figure 6. Missing comparison with controls (WT and virus with GFP)
We have modified the figure 6 to replot the WT control (black dashed lines) from the figures 3-5. Since the AAV with GFP only was only injected in behavioral experiments, and since the seizure activity in mock-injected experiments was not reduced compared to Syn2 KO (Fig. 5 B), we did not replot this control in figure 6.

Reviewer 2 Report
Major issues
#1. Even though I am interested in this paper, the purpose of the study, hypothesis have not been written. Therefore, still some revisions are needed before publication.
Since I found the purposes in the context, please clearly state the hypothesis, purpose, primary and secondary outcomes of this study
#2. Could you show some figures to explain the mechanism?
Minot issues
#1. The word of NMDA needs full words.
Author Response
#1. Even though I am interested in this paper, the purpose of the study, hypothesis have not been written. Therefore, still some revisions are needed before publication.
Since I found the purposes in the context, please clearly state the hypothesis, purpose, primary and secondary outcomes of this study
To address these questions, we completely rewrote the Discussion section of the revised manuscript. We now clearly state that the major questions we asked were: 1) what are the contributions of the direct Syn II action in the synapse versus developmental compensatory mechanisms in the Syn II deleted brain; and 2) what are the molecular mechanisms underlying the overexcitability in Syn2 KO brain? We also stated our main outcomes: 1) Syn II in the intact brain in vivo directly suppresses the excitability in neuronal networks and behavioral epileptic seizures; and 2) the PKA/CaMKI phosphorylation of Syn II contributes to the regulation of epilepsy predisposition, even though this mechanism does not apparently contribute to the regulation of spontaneous glutamatergic transmission.
#2. Could you show some figures to explain the mechanism?
In the revised manuscript, we included the new figure (6 A), which shows Syn II phosphorylation site and illustrated its presumed function
Minot issues
#1. The word of NMDA needs full words.
This is spelled out in the revised Methods section

Round 2
Reviewer 1 Report
The authors answered my questions.
Two minor comments:
- lines 89-90. Something wrong with the composition of the solution. Please check.
- Figure 1A. Please add information about staining protocol with a fluorescent 181 Nissl (Neurotrace 500/525) to Methods.
Author Response
- lines 89-90. Something wrong with the composition of the solution. Please check.
Thank you for pinpointing the error. It has been corrected in the revised manuscript, lines 88-90
2. Figure 1A. Please add information about staining protocol with a fluorescent 181 Nissl (Neurotrace 500/525) to Methods.
This description has been added to the revised Methods section, lines 129-132
Reviewer 2 Report
The authors revised in accordance with my comments. I endorse this version.
Author Response
Thank you for reviewing our manuscript